# Therapeutic Effects of a Novel Aptamer on Coronaviral Infection-Induced Lung Injury and Systemic Inflammatory Responses

**DOI:** 10.3390/cells13050422

**Published:** 2024-02-28

**Authors:** Yingchun Wang, Mikael Lindstam, David Hwang, Luiza Jedlina, Mingyao Liu

**Affiliations:** 1Latner Thoracic Surgery Research Laboratories, Toronto General Hospital Research Institute, University Health Network, Toronto, ON M5G 1L7, Canada; yingchun.wang2@uhn.ca; 2Aptahem AB, 211 22 Malmo, Sweden; ml@aptahem.com (M.L.); lj@aptahem.com (L.J.); 3Department of Laboratory Medicine and Pathobiology, Temerty Faculty of Medicine, University of Toronto, Toronto, ON M5S 1A8, Canada; david.hwang@utoronto.ca; 4Departments of Surgery, Medicine, and Physiology, Institute of Medical Science, Temerty Faculty of Medicine, University of Toronto, Toronto, ON M5S 1A8, Canada

**Keywords:** Coronaviral infection, Aptamer, anti-inflammatory response, acute lung injury

## Abstract

Background: Coronaviral infection-induced acute lung injury has become a major threat to public health, especially through the ongoing pandemic of COVID-19. Apta-1 is a newly discovered Aptamer that has anti-inflammatory effects on systemic septic responses. The therapeutic effects of Apta-1 on coronaviral infection-induced acute lung injury and systemic responses were evaluated in the present study. Methods: Female A/J mice (at 12–14 weeks of age) were challenged with murine hepatitis virus 1 (MHV-1), a coronavirus, at 5000 PFU intranasally, followed by Apta-1 intravenously administered (100 mg/kg, twice) 1.5 h or 2 days after viral delivery. Animals were sacrificed at Day 2 or Day 4. Lung tissues were examined with H&E, immunohistochemistry staining, and western blotting. RT-qPCR was used for cytokine gene expression. Serum and plasma were collected for laboratory assessments. Results: Apta-1 treatment reduced viral titers, prevented MHV-1-induced reduction of circulating blood volume and hemolysis, reduced alveolar space hemorrhage, and protease-activated receptor 1 (PAR-1) cleavage. Apta-1 treatment also significantly reduced chemokine (MKC, MCP-1, and RANTES) levels, as well as AST, ALT, total bilirubin, and reduced unconjugated bilirubin levels in the serum. Conclusion: Apta-1 showed therapeutic benefits in coronaviral infection-induced hemorrhage and PAR-1 cleavage in the lung. It also has anti-inflammatory effects systemically.

## 1. Introduction

Coronavirus family members can infect a wide range of animal species in nature. Three of them crossed the species barrier, attacking the human lower respiratory tract, resulting in a notable outbreak of SARS in 2003, MERS in 2012, and COVID-19 currently. COVID-19, caused by the SARS-CoV2 viral infection, can lead to a series of clinical situations from non-symptomatic viral carriers/spreaders to severe illness characterized by acute respiratory distress syndrome (ARDS), multi organ failure, and death [1,2]. In COVID-19, the lack of specific and effective therapeutic agents is one of the major challenges of dealing with patients, especially those who developed ARDS. This has promoted studies on exploring new therapeutics on treating coronaviral infection-induced acute lung injury both experimentally and clinically.

Depending on the mouse strain and viral strain and dosage, murine coronaviral infection could be lethal, acute, severe, or sub-chronic [3,4]. By screening 4 mouse strains against 5 coronavirus strains, it was found that female A/J mice challenged with murine hepatitis virus 1 (MHV-1) induced acute lung injury [3]. The MHV-1 virus also induced lung injury and fibrosis in older C57BL6/J mice [5]. Utilizing these models, it has been found that proteasome inhibition promoted animal survival [6], and long pentraxin PTX3 administration attenuated lung injury and fibrosis [5]. Pathological features observed in this model are shared between SARS and COVID-19: induced by coronavirus, severe injury in the lung, associated with over expression of cytokines and chemokines, disorders in blood coagulation, etc.

Aptamers are oligonucleotide sequences with a length of about 25–80 bases. They generally fold into diverse three-dimensional structures that bind to specific targets. They can bind and interfere with the function of a target protein [7]. Aptamers have been explored as therapeutic agents, for example, Pegaptanib is an aptamer that has been approved by the FDA for clinical use [8]. Aptamers have been studied for various microbial infections [9] including viral infections [10]. The Apta-1 aptamer is an RNA oligonucleotide with a high affinity and selectivity for the heparin-binding motif (Exosite II) on thrombin, thereby preventing the cleavage of PAR-1 and PAR-4 receptors [11]. It has been shown to inhibit LPS-induced inflammation in mice and non-human primates [12,13].

We hypothesized that after coronavirus infection, the administration of Apta-1 may inhibit acute inflammation and subsequently reduce acute lung injury and systemic inflammatory responses. In the present study, we first used an LPS administration model to test the local therapeutic effects of Apta-1 in murine lungs, and then used the MHV-1 virus to test the effects of Apta-1 on coronaviral infection-induced acute lung injury and systemic inflammatory responses.

## 2. Materials and Methods

### 2.1. Animals

Female C57BL/6J mice and A/J mice were purchased from Jackson Laboratory (Bar Harbor, ME, USA). All procedures carried out in mice were approved by the Animal Use and Care Committee of the University Health Network. All animals received humane care in compliance with the Guide for the Care and Use of Experimental Animals formulated by the Canadian Council on Animal Care. Animals were housed in a temperature- (22 ± 2 °C) and light-controlled (12 h light/12 h dark cycle) facility. Body weight changes and clinical signs including ruffled fur, increased respiratory rate, tremors, lack of activity, and survival were monitored and recorded daily [14].

### 2.2. LPS-Induced Acute Lung Injury

Six-week-old female C57BL/6J mice were administrated with a dose of 10 mg/kg of LPS (*Escherichia coli* 0111:B4, Sigma, St. Louis, MO, USA) (1 mg/mL in PBS) via an intraperitoneal injection. Two doses of Apta-1 (Aptahem AB, Malmo, Sweden) (20 mg/mL in 0.9% saline, 5 mL/kg, 100 mg/kg) were administrated at 15 min and 60 min via an intravenous (iv) injection after LPS administration [12]. A subgroup of LPS administrated mice were injected twice with 0.9% saline, instead of Apta-1, as a vehicle control. As a negative control, another group of mice were injected with PBS (instead of LPS) intraperitoneally, and then injected twice with 0.9% saline intravenously (5 mL/kg). All animals were sacrificed at 28 h after LPS or PBS administration. Lung tissue and blood were collected at the end of experimentation.

### 2.3. Viral Challenge and Apta-1 Treatment

The MHV-1 virus was obtained from the American Type Culture Collection (Manassas, VA, USA). Viruses were first plaque purified and then expanded in murine 17CL1 cells. Supernatants were collected and stored at −80 °C [3]. For the dose-dependent body weight curve study, 12–14 week-old female A/J mice were anesthetised using an intraperitoneal injection with pentobarbital followed by intranasal inoculation of the MHV-1 virus at different doses [5]. Due to its narrow safe working range, pentobarbital was replaced by Ketamine/Xylazine in the Apta-1 treatment studies. After anesthesia, two doses of Apta-1 (100 mg/kg) were iv injected at 1 h intervals, either at 1.5 h (Day 0) or 48 h (Day 2) after MHV-1 inoculation. Apta-1 was replaced with 0.9% saline in the MHV-1 only group. In some studies, a control group of mice were intranasally inoculated with DMEM (the vehicle used for MHV-1 viral delivery) and then iv injected with 0.9% saline, as a non-infection non-treatment control. Mice were sacrificed at Day 2 or Day 4 post-infection.

### 2.4. Viral Titers

The post-caval lobe of the lung was homogenized in ice-cold DMEM in a ratio of 1:10, utilizing a Tissuelyser LT (Qiagen, Toronto, ON, Canada) for 4 min at 20 Hz, and centrifuged at 4 °C for 10 min at 3000 rpm. The supernatant was stored at −80 °C. The virus titer was determined with the Plaque assay. Briefly, L2 cells were seeded into 6-well plates with DMEM + 10% FBS. When cells reached 80% confluence, cells were infected with tissue supernatant at a serial dilution from 10^−1^ to 10^−6^. An agar plaque assay was performed to determine the virus titer, as described previously [5].

### 2.5. Assessment of Lung Injury

When the animal was sacrificed, body weight was recorded, and lung tissues were collected. The whole left lung was fixed in 10% formalin for 72 h for histology and immunohistochemistry (IHC) studies. H&E staining of the lung tissue was used for assessing alveolar damage. For deflated lungs, 3 randomly chosen fields at 20× were analyzed in a blind fashion by three investigators. Five parameters including neutrophil infiltration, hyaline membrane, alveolar septal thickening, alveolar hemorrhage, and cellular hyperplasia were assessed.

In a subgroup, mouse lung tissues were perfused with heparinized PBS and inflated with formalin and then fixed in formalin. For these inflated and perfused lungs, H&E sections were scored by a pathologist in a blinded fashion to assess: (1) air space haemorrhage, (2) vascular congestion, (3) edema/fibrin in the alveoli, and (4) presence of infiltrating white blood cells. These criteria were graded on a scale ranging from normal appearance (0%), mild (<10%), moderate (10–50%), and severe (>50%) abnormalities and scored from 0 to 3, respectively.

### 2.6. Immunohistochemistry Staining

Infiltration of neutrophils, macrophages, and lymphocytes were examined with IHC staining in deflated lungs [3]. The antibodies used were for detecting T-cells (anti-CD3, #C7930, Sigma), B-cells (anti-B220, #550286, BD Biosciences, Mississauga, ON, Canada), neutrophils (anti-Ly-6G, #87048, Cell Signaling Technology, Danvers, MA, USA), and macrophages (anti-F4/80, #70076, Cell Signaling Technology). The IHC positively stained cells were quantified using Halo image analysis platform (Indica Labs, Albuquerque, NM, USA).

### 2.7. Quantitative Reverse Transcription PCR (RT-qPCR)

The superior and the middle lobe of the right lung were used for RT-qPCR to determine cytokine/chemokine gene expression. Briefly, 20–30 mg lung tissue was homogenized in 600 μL RLT buffer, using a Tissuelyser II. Total RNA was isolated using the RNeasy mini kit (Qiagen). The cDNA was generated by the iScript cDNA Synthesis Kit (Bio-Rad, Hercules, CA, USA). Real-time PCR was performed using the SsoAdvanced™ Universal SYBR^®^ Green Supermix (Bio-Rad) with a 7900 fast real-time PCR system (Applied Biosystems, Waltham, MA, USA). For each gene tested, two primer sets were used to rule out artifacts. Primer efficiency was measured against a standard curve, which was generated using a serially diluted cDNA template. For each PCR reaction, two technical replicates were run and an average Cq was used to calculate the fold change using the Pfaffl method with Cq < 40 as the cut-off. *Ppib* was used as reference gene for quantification normalization. The RT-qPCR primers used are listed in Appendix A.

### 2.8. Western Blotting

The upper right lung lobes were homogenized utilizing a Tissuelyser LT in RIPA lysis buffer supplemented with a mixture of protease inhibitors cocktail (Roche Applied Science, Indianapolis, IN, USA) and phosphatase inhibitors. The homogenate was rotated for 2 h at 4 °C. The lysate was centrifuged for 20 min at 12,000 rpm, and the supernatant was collected. Protein concentrations were measured using the Pierce^TM^ BCA Protein assay Kit (Thermo Fisher Scientific, Rockford, IL, USA). Solubilized proteins were denatured in Laemmli buffer and separated utilizing SDS-PAGE. Rabbit anti-PAR1 (Bioss, BS-0828R) and rabbit anti-thrombin (Novus biologicals, NBP1-58268-20 mL, Toronto, ON, Canada) antibodies were probed, using the anti-GAPDH antibody as a loading control. Chemiluminescence signals were captured with the ChemiDoc MP Imaging System (Bio-Rad) and the density was analyzed using image lab software (latest v. 5.0 Bio-Rad).

### 2.9. Serum and Plasma Collection and Tests

For serum preparation, blood was collected using a 25 G gauge needle through cardiac puncture. Serum was prepared following the instructions from IDEXX Bioanalytics (North Grafton, MA, USA). Serum creatinine, blood urea nitrogen (BUN), aspartate aminotransferase (AST), alanine aminotransferase (ALT), alkaline phosphatase, total protein, albumin, and globulin levels were measured by IDEXX Bioanalytics. Cytokine levels in serum were quantified by IDEXX Bioanalytics using the Milliplex MAP Mouse Cytokine/Chemokine Magnetic Bead Panel (MCYTOMAG-70K-PMX, MilliporeSigma, Burlington, MA, USA).

For plasma preparation, blood was collected using a 1 mL syringe with a 25 G gauge needle containing 50 μL of 3.8% sodium citrate through the caudal (inferior) vena cava. Collected blood was transferred immediately into a microcentrifuge tube. Extra 3.8% sodium citrate was added to reach a final concentration of 0.38% in blood preparation. Citrated blood was centrifuged at 1500 g for 20 min and the supernatant was collected and stored at −80 °C [15]. Prothrombin Time and Partial Thromboplastin Time in the citrated plasma were measured by IDEXX Bioanalytics (Coagulation Mini Panel).

### 2.10. Statistical Analysis

Unless indicated otherwise, statistical analysis was performed using the unpaired, two-tailed Student’s *t*-test with GraphPad Prism 9.0 (GraphPad, La Jolla, CA, USA). Clinical biochemistry, cytokines in the serum, and PAR-1 Western blot data were analyzed using two-way ANOVA followed by the multiple unpaired *t*-test. Lung injury score data were analyzed using two-way ANOVA, or one-way ANOVA, followed by the Mann–Whitney test. Hemolysis was analyzed using the Mann–Whitney test. The RT-qPCR of the LPS model was analyzed using one-way ANOVA followed by the unpaired *t*-test. Plasma Prothrombin Time and Partial Thromboplastin Time were analyzed using one-way ANOVA. All data are presented as means ± standard error of the mean (SEM). *p* < 0.05 is considered statistically significant.

## 3. Results

### 3.1. Apta-1 Reduces Airway LPS-Induced Acute Lung Injury

Apta-1 treatment reduced LPS-induced systemic septic responses in mice and in monkeys [12,13]. To determine the local therapeutic effects of Apta-1 in the lung tissue, LPS (10 mg/kg) was administrated to female C57BL/6J mice intraperitoneally, followed by intravenous injections of Apta-1 after 15 min and 60 min (100 mg/kg) (Figure 1A). This strategy was optimized previously in murine studies to maximize the Apta-1 delivery [12]. The baseline body weights of mice before the LPS challenge were 18.91 ± 1.32 g (mean ± SD), and no differences among the 3 groups were observed. After 24 h, the LPS-treated animals showed significant body weight loss and reduced blood volume in circulation. In mice who received the Apta-1 treatment, body weight loss was further increased (Appendix A); however, the blood volume collected was significantly higher in the Apta-1 treated animals (Appendix A).

The gene expression of cytokines and chemokines in the lung tissue was analyzed with RT-qPCR. The Apta-1 treatment significantly reduced the LPS-induced elevation of mRNA levels of *Ip-10, Ptx3*, and *Il-6* (Figure 1B–D). Similar effects have been shown by other potential therapeutics in LPS-induced acute lung injury [16,17,18], suggesting an anti-inflammatory effect of Apta-1 in the lung tissue. IP-10 neutralization ameliorated LPS-induced acute lung injury [19]. IL-6 has been recognized as a biomarker for ARDS [20]. PTX3 has also been proposed as a biomarker for pulmonary infection and acute lung injury [16,21]. The mRNA levels of *Mip-1α, Il-10*, and *Il-12α* were also reduced in the Apta-1 treated group but did not reach statistical significance (Figure 1E–G). Apta-1 treatment has no effects on LPS-induced elevation of *Il-1β, Tnf-α*, and *Mcp-1* (Appendix A). IL-1β and TNF-α are early response cytokines [22], and their gene expression may have passed the peak expression. They are more useful as markers for the severity of sepsis rather than ARDS [20]. Interestingly, airway LPS-treatment reduced mRNA levels of *Il-33*, which was not affected by Apta-1 (Appendix A). These results confirmed the local anti-inflammatory effects of Apta-1. The dose, route, and regimen of Apta-1 delivery were used in the subsequent MHV-1 studies.

### 3.2. Apta-1 Treatment Reduces MHV-1 Coronaviral Titer, Loss of Blood Volume, and Hemolysis

Different from the LPS-induced pathology, viral-infection has a latent phase for the virus to replicate. After the viral particles accumulate to a certain level, an inflammatory response is induced, leading to cytokine release, acute lung injury, disorder in the immune system, and coagulation. To find this pathological process, a time-course study was conducted first. To determine the efficacy of the MHV-1 viral infection, female A/J mice around 12–14 weeks of age were challenged utilizing an intranasal inoculation of 500, 1000, or 5000 PFU. Body weight change was monitored while clinical signs were recorded as described in our previous studies [14]. At the two lower viral doses, the animals’ body weights showed fluctuations, while the body weights of mice challenged with 5000 PFU continued to decrease in the first 6 days and then returned afterwards (Figure 2A). We then used this dose for subsequent studies.

To determine the effects of Apta-1 on the MHV-1 viral infection in A/J mice, two treatment regimens were tested. Animals were initially challenged with MHV-1. To determine if Apta-1 could treat the MHV-1 viral infection-induced lung injury, in the first group, animals received Apta-1 (100 mg/kg) twice intravenously on Day 2 after viral infection. Animals were sacrificed on Day 4 (D2D4). To determine if an earlier treatment could prevent the progress of the viral infection-induced lung injury, Apta-1 was given immediately on Day 0, and the animal was sacrificed at Day 2 (D0D2), or at Day 4 (D0D4) (Figure 2B).

The viral titers in the lung tissues significantly decreased from Day 2 to Day 4 post-infection, and Apta-1 treatment significantly reduced the viral titers in the D2D4 group (Figure 2C). The baseline body weights of mice before viral infection were 21.02 ± 1.67 g (mean ± SD), with no difference among subgroups. The MHV-1 virus induced significant body weight loss from Day 2 to Day 4, which was not affected by Apta-1 treatment (Figure 2D). Significantly higher serum volumes were collected at Day 2 and Day 4 from animals treated with Apta-1 at Day 0 (Figure 2E). At Day 4, the MHV-1 viral infection induced hemolysis, which was significantly reduced by theApta-1 treatment, especially in the D0D4 group (Figure 2F). In summary, the Apta-1 treatment showed systemic benefits in the MHV-1 viral infected animals.

### 3.3. Apta-1 Treatment Reduced Alveolar Hemorrhage and Cleavage of PAR-1

An intratracheal challenge of mice with MHV-1 induced acute lung injury was assessed with H&E staining on lung tissue slides. The lung tissues were immersed in formalin without inflation. At low magnification, the histology of the MHV-1 viral infected lungs showed atelectasis and hemorrhage under the pleural membrane, which were less apparent in the Apta-1 treated group (Appendix A). Blinded lung injury scoring at higher magnification also showed significantly lower hemorrhage in the lung tissue of the Apta-1 treated groups (Figure 3A). To confirm the lung injury score results, we repeated D0D2 groups with inflation perfusion fixation. Again, the Apta-1 treatment reduced hemorrhage in the alveolar space (Figure 3B,C). However, fibrin deposition, vascular congestion, leukocyte infiltration, and total lung injury score did not show statistical significance (Appendix A). The infiltration of macrophages, neutrophils, T cells, and B cells was examined with immunohistochemistry staining and scored semi-quantitatively. No statistical differences were found (Appendix A).

The selective inhibition of Apta-1 on pulmonary hemorrhage is interesting, as cross talks between inflammation, coagulation, and pulmonary endothelial integrity has been considered as a major mechanism for acute lung injury, and PAR-1 activation plays an important role on endothelial permeability [23]. Anti-PAR-1 western blotting shows two bands around 53 KDa, representing the full length and cleaved PAR-1, respectively. In the normal (no administration, NA) and negative control (CTL) samples, the top PAR-1 protein bands were clearly dominant. In the MHV-1 treated samples, the cleaved band became dominant, and the density of the top bands not only significantly reduced, but also shifted towards the lower bands, indicating an active cleavage of PAR-1 (Figure 4A). We quantified the ratio of the density between the lower bands and the top band, Apta-1 significantly reduced the ratio (Figure 4B). We also conducted western blotting on samples collected from the initial D0D2, D0D4, and D2D4 groups. The density ratio of these two bands significantly increased in Day 4 samples, and the Apta-1 treatment significantly reduced the ratio (Figure 4C). The protein levels of thrombin were not altered by the Apta-1 treatment. These results indicate that Apta-1 may reduce PAR-1 cleavage and activation and partially protect pulmonary endothelial integrity.

The expression of cytokine genes in lung tissues was tested with RT-qPCR. Among the genes (*Ip-10, Il-6, Ptx3, Mip-1a, Il-10, Il-12a, Il-1b, Tnfa, Mcp-1, Il-33, Il-7, Il-18, Il-2*) tested, only the *Il-6* gene was significantly up-regulated by the Apta-1 treatment in the D0D4 samples (MHV-1: 1.05 ± 0.11 vs. MHV-1 + Apta-1: 1.66 ± 0.19, *p* < 0.05, ratio with housekeeping gene, mean ± SE). IL-6 is a pleiotropic cytokine with both pro- and anti-inflammatory properties [24]. Even though the IL-6 gene and protein levels have been used as biomarkers for inflammatory responses [20], the protective effects of IL-6 have been reported in acute lung injury [25,26].

### 3.4. Apta-1 Treatment Reduced MHV-1 Viral Infection-Induced Systemic Inflammatory Responses

To determine the effects of Apta-1 treatment on MHV-1 viral infection-induced systemic responses, the serum cytokine concentrations were measured with a multiplex bead assay. The levels of IL-6, IFNγ, IP-10, were significantly decreased at day 4 after viral infection, but not affected by Apta-1 treatment (Figure 5A–C). The serum levels of MKC, MCP-1, and RANTES were significantly lower in the animals treated with Apta-1 (Figure 5D–F). MKC (murine KC, CXCL1), MCP-1 (CCL2) and RANTES (CCL5) are chemokines that are involved in the pathogenesis of acute lung injury [27]. The levels of MIP-1β, IL-1α, G-CSF, and IL-9 were affected neither by MHV-1 viral infection nor Apta-1 treatment (Appendix A).

To determine the systemic effects of the Apta-1 treatment, the serum samples collected were also used for clinical biochemistry analyses. The serum levels of AST, ALT, total bilirubin, and unconjugated bilirubin were significantly increased from Day 2 to Day 4 (Figure 6). In the D0D4 samples, the serum AST levels were significantly lower in the Apta-1 group (Figure 6A), and the ALT levels were also lower (*p* = 0.10, Figure 6B). The total bilirubin and unconjugated-bilirubin levels were significantly lower in the Apta-1 treated animals (Figure 6C,D). The laboratory findings of COVID-19 patients showed that ICU patients had significantly higher levels of AST, ALT, and total bilirubin in circulation [1]. Reducing these levels may be clinically beneficial.

The Apta-1 treatment did not affect BUN, alkaline phosphate (ALP), creatine kinase, total protein, albumin, globulin, albumin/globulin ratio, glucose, and cholesterol levels (Appendix A). The Apta-1 did not affect Na^+^ levels, but significantly reduced K^+^ levels, and therefore significantly increased the Na^+^/K^+^ ratio (Appendix A). Other electrolyte levels in the serum (phosphorus, Ca^++^, Cl^-,^ and bicarbonate) (Appendix A) and coagulation related measures (prothrombin time and partial thromboplastin time) in plasma (Appendix A) were not affected by the Apta-1 treatment.

## 4. Discussion

### 4.1. Animal Models for Corona Viral Infection

The recent outbreak of COVID-19 caused by SARS-CoV-2, together with SARS-CoV and MERS-CoV, raised urgent calls to study the underlying mechanisms caused by coronaviruses and to explore potential therapeutics with animal models [28]. The invasion of SARS-CoV-2 relies on its spike protein binding to the ACE2 receptor on host cells via a receptor-binding domain [29]. Therefore, human ACE2 transgenic mice have been used as a model for COVID-19 [30]. These mice are restricted to a single genetic background. To overcome this limitation, Hassan et al. transduced replication-defective adenoviruses encoding human ACE2 via intranasal administration into BALB/c mice and established the hACE2 expression in lung tissues, and then infected these mice with SARS-CoV-2 [31]. Similarly, Israelow et al. developed a mouse model based on the adeno-associated virus-mediated expression of hACE2 [32]. These new mouse models allow for the use of mice from different genetic backgrounds. However, the SARS-CoV-2 variants used in these studies require level 3 facilities for safety reasons. To determine whether Apta-1 has therapeutic benefits for coronaviral infection-induced acute lung injury, we choose to use the MHV-1 coronaviral infection in female A/J mice developed previously [3], within regular level 2 facilities. This cannot be claimed as an animal model for COVID-19 but it provides knowledge on the coronaviral infection in the lung and resulting systemic responses.

In the present study, we first tested the doses and timing of MHV-1 induced inflammatory responses. Clinically, most of the severe COVID-19 patients are old adults [1]. Therefore, we used mice at 12–14 weeks of age, which are equivalent to mature humans. Similar to our previous report, the MHV-1 airway delivery resulted in systemic responses and body weight loss, only one animal died after receiving 5000 PFU virus, and all other animals recovered after one-week. The circulating cytokine (IL-6, IFNγ, and IP-10) levels were decreased at Day 4 after infection. These responses are less severe than what has reported in the literature [3], which may be due to differences in the preparation of the viruses, and older A/J mice used in the present study. Nonetheless, these responses are very similar to the human SARS-CoV-2 infection-induced responses in other mouse models [30,31,32].

### 4.2. Apta-1, a Novel Anti-Inflammatory Aptamer

Apta-1 was first discovered as an RNA aptamer against a conserved region of the Plasmodium falciparum erythrocyte membrane protein 1 [33]. Later, it was shown to inhibit LPS-induced systemic septic responses in mice and non-human primates [12,13]. In the present study, the intravenous administration of Apta-1 effectively reduced gene expression of several cytokines and chemokines induced by LPS in the lung. Not only does this experiment support the anti-inflammatory benefits of Apta-1 in acute lung injury, but also demonstrates that the dose and delivery route of Apta-1 used were effective, which were then also used in subsequent studies. Interestingly, in the MHV-1 viral infected AJ mice, Apta-1 did not show significant effects on cytokine gene expression in the lung at the time of sample collection. The inflammatory responses induced by LPS or the MHV-1 virus are mediated through different signal transduction pathways. Moreover, the responses are affected by the doses, timing, animal strains, and many other co-founding factors. The effects of Apta-1 on the MHV-1 virus-induced inflammatory response in the lung should be further investigated.

### 4.3. Apta-1 Reduces Coronaviral Infection-Induced Pulmonary Hemorrhage and PAR-1 Cleavage

The Apta-1 treatment did not reduce the total lung injury score, fibrin deposition, vascular congestion, and leukocyte recruitment in the lung. However, the pulmonary hemorrhage was significantly reduced by Apta-1 in the lung. Cross talks between inflammation and coagulation have been recognized to be a major underlying mechanism for acute lung injury and its severe form, ARDS. The activation of PAR-1 by thrombin may up-regulate inflammatory genes in the lung and disrupt the endothelial barrier function [23]. *Par*-1 gene knock out affords protection to mice from bleomycin-induced lung inflammation and fibrosis [34]. The PAR-1 antagonist protects against ischemia reperfusion-induced lung injury [35]. PAR-1 has been proposed as a potential therapeutic target for COVID-19 [36].

PAR-1 contains a hirudin-like domain, which has a high affinity thrombin binding site to recruit thrombin and enable thrombin to activate PAR-1 specifically and efficiently [37]. In the present study, we do not have evidence indicating that the cleavage of PAR-1 is mainly via thrombin. In fact, the top bands of PAR-1 in the western blots were smaller (shifted down from its original molecular weight), indicating that the top bands in the viral infected lung tissues could also be proteolytic products. PARs can be specifically cleaved and irreversibly activated by various proteases [37]; therefore, we speculate that the MHV-1 inflection may lead to the activation of multiple proteases, and that the inhibited cleavage of PAR-1 by Apta-1 may help to reduce hemorrhage in the lung tissue.

### 4.4. Apta-1 Reduces Coronaviral Infection-Induced Systemic Inflammatory Responses

Compared with the limited therapeutic effects observed in the lung tissue, the Apta-1 treatment reduced the MHV-1 coronaviral infection-induced systemic responses more effectively. The Apta-1 treatment reduced viral titers in the lung tissue, prevented the reduction of circulating blood volume and hemolysis, and reduced chemokines in the serum. Clinically, COVID-19 ICU patients had higher plasma levels of multiple cytokines and chemokines [1]. Lower chemokine levels may indicate reduced inflammatory responses. The laboratory findings of COVID-19 patients showed that ICU patients had significantly higher levels of AST, ALT, and total bilirubin in circulation [1]. The Apta-1 treatment significantly reduced these levels. Since MHV-1 is more effective in inducing hepatitis in mice, we speculate that an airway delivered MHV-1 virus may gradually damage liver function, which was at least partially protected by the Apta-1 treatment. In the present study, Apta-1 was administrated intravenously after viral infection, especially one group of animals received Apta-1 two days after the MHV-1 administration. Therefore, these positive results can be considered as therapeutic benefits.

### 4.5. Limitations of the Present Study

Due to the limited therapeutic effects of Apta-1 observed in the lung tissue, we did not further study the functional endpoints of acute lung injury such as lung mechanics, microvascular permeability, lung wet/dry weight, blood oxygenation, heart rate, breathing rate, etc. Based on previous reports [3], only older female AJ mice were used in the MHV-1 viral infection study. The incidence of COVID-19 is similar between men and women, but the severity and mortality are much worse in men [10,38]. Male mice were more susceptible to SARS-CoV infection compared to age-matched females [39]. The effects of MHV-1 and Apta-1 on male animals should be tested. Based on the results from the LPS-induced acute lung injury [12], we only used one regimen of Apta-1 in the MHV-1 studies and the time points examined were limited. The longitudinal studies were not conducted. Further investigations are necessary to reveal the effects of Apta-1 in coronaviral infection. Ideally, we should add a normal control and an Apta-1 only group for comparison. However, the cost of AJ mice is very high. In our previous studies, we did not observe reportable biological effects induced by Apta-1. In our previous toxicological studies, in which mice, rats, and non-human primates were given various doses of Apta-1, no adverse effects were detected [12,13].

## 5. Conclusions

Results from the present study indicate that Apta-1, an aptamer, has therapeutic effects on coronaviral infection-induced acute lung injury and systemic inflammatory responses. The underlying mechanisms and its clinical application in coronaviral infection should be further explored.

## Figures and Tables

**Figure 1 cells-13-00422-f001:**
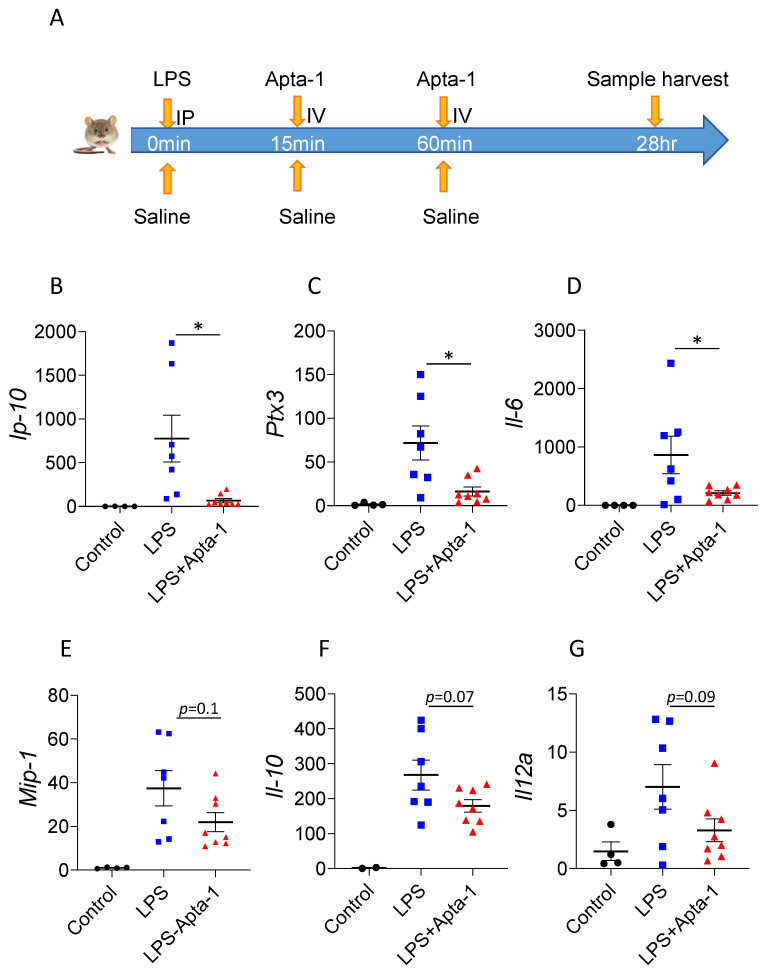
Apta-1 inhibits LPS-induced acute inflammatory responses in mouse lungs. (**A**). Experimental design. Female C57BL/6J mice were challenged intraperitoneally with LPS (10 mg/kg), followed by an intravenous injection of Apta-1 (100 mg/kg, twice). One day later, animals were sacrificed, and gene expression of cytokines in the lung was analyzed using RT-qPCR. Apta-1 treatment significantly inhibited LPS-induced gene expression of *Ip-10* (**B**), *Ptx3* (**C**), *Il-6* (**D**), and reduced the expression of *Mip-1* (**E**), *Il-10* (**F**), and *Il-12a* (**G**), although it did not reach statistical significance. Expression of the gene of interest was normalized with housekeeping gene *Ppi1b* and expressed as the ratio with the average of the control group. *: *p* < 0.05. One-way ANOVA followed by the unpaired *t*-test. The number of mice: 4 for control, 7 for LPS, and 8 for LPS plus Apta-1.

**Figure 2 cells-13-00422-f002:**
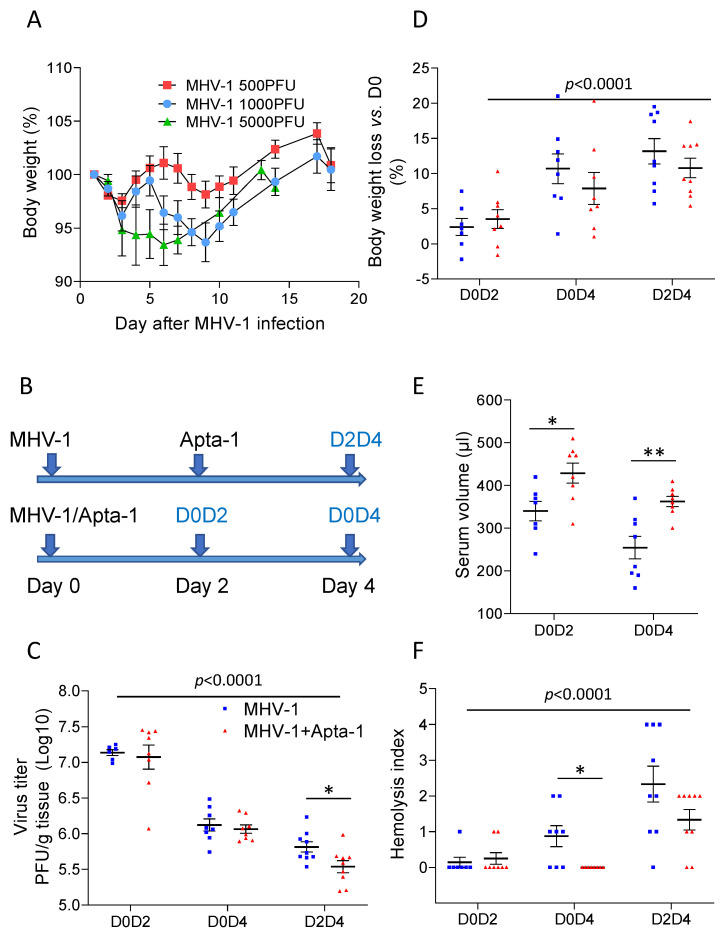
Apta-1 reduces the MHV-1 coronaviral titer, viral infection-induced loss of circulating blood, and hemolysis. (**A**). Dose responses of MHV-1 on body weight changes in mice. Female A/J mice at 12-14 weeks of age were challenged intratracheally with the MHV-1 coronavirus at 500 PFU (n = 6), 1000 PFU (n = 6), or 5000 PFU (n = 3). At 5000 PFU, MHV-1 induced more body weight lose with delayed recovery. This dose was used in the subsequent studies. (**B**). Experimental design. After intranasal delivery of the MHV-1 coronavirus, A/J mice received the Apta-1 treatment (100 mg/kg, twice) at Day 2 and were sacrificed at Day 4 (D2D4) or received the Apta-1 at Day 0 and were sacrificed at Day 2 (D0D2) or Day 4 (D0D4). Mice in control groups received an equal volume of vehicle solution. (**C**). The MHV-1 viral titer was significantly lower at Day 4 (*p* < 0.0001). Apta-1 treatment significantly reduced the viral titer in the D2D4 group. (**D**). The MHV-1 viral infection induced more body weight loss at Day 4 (*p* < 0.0001). Apta-1 did not affect the MHV-1 induced body weight changes. (**E**). The serum volumes collected from the D0D2 and D0D4 groups were significantly higher in the Apta-1 treated animals (*t*-test). (**F**). The MHV-1 viral infection induced more hemolysis at Day 4 (*p* < 0.0001). The Apta-1 treatment significantly reduced hemolysis in the D0D4 group. Two-way ANOVA followed by the Mann–Whitney test. *: *p* < 0.05; **: *p* < 0.01. Numbers of mice in MHV-1 and MHV-1 + Apta-1 groups are: 7 and 8 in D0D2 groups; 8 and 8 in D0D4 groups; 9 and 9 in D2D4 groups.

**Figure 3 cells-13-00422-f003:**
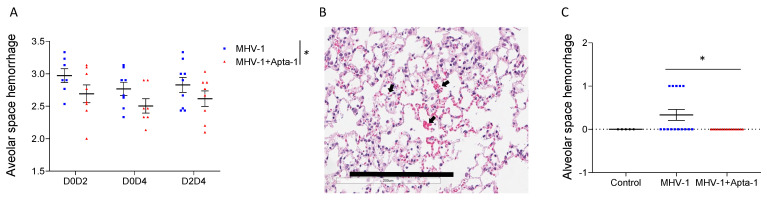
Apta-1 treatment reduced alveolar space hemorrhage. The lung injury score was blindly evaluated by a pathologist. (**A**). Alveolar space hemorrhage was significantly less in the Apta-1 treated groups than in the MHV-1 only groups. The vertical line highlights the two experimental conditions. *: *p* < 0.05; two-way ANOVA. Numbers of mice in MHV-1 and MHV-1 + Apta-1 groups are: 7 and 8 in D0D2 groups; 8 and 7 in D0D4 groups; 9 and 8 in D2D4 groups. (**B**) and (**C**). D0D2 groups were repeated with inflation perfusion fixation. (**B**). Alveolar space hemorrhage (indicated with arrows) is seen in the MHV-1 group. The scale bar is 200 μm. (**C**). The Apta-1 treatment prevented alveolar space hemorrhage (one-way ANOVA followed by the Mann–Whitney test). Numbers of mice in control, MHV-1, and MHV-1 + Apta-1 groups are: 5, 15, and 15, respectively.

**Figure 4 cells-13-00422-f004:**
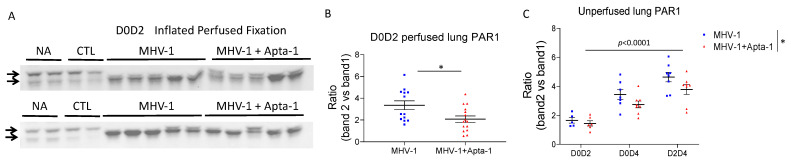
Apta-1 treatment significantly reduced PAR-1 cleavage in the lung tissue of MHV-1 viral infected mice. (**A**). Two sets of original western blots as examples show two bands of PAR-1 proteins. In the MHV-1 viral infected lung tissue, the lower bands became dominate and the top bands were shifted towards the lower bands. NA: no administration; CTL: control animals inoculated with DMEM instead of MHV-1 as a vehicle control and then treated with saline instead of Apta-1. (**B**). In inflated perfused lung tissue, the ratio of lower bands to higher bands was significantly lower in the Apta-1 treated lungs (*t*-test). Numbers of mice in MHV-1 and MHV-1 + Apta-1 groups are: 13 and 15, respectively. (**C**). In non-perfused lungs, the ratio was significantly increased at Day 4, and it was significantly lower in the Apta-1 treated groups (two-way ANOVA). The vertical line highlights the two experimental conditions. *: *p* < 0.05. Numbers of mice in MHV-1 and MHV-1 + Apta-1 groups are: 5 and 6 in D0D2 groups; 7 and 8 in D0D4 groups; 8 and 8 in D2D4 groups. Equal loading was confirmed with GAPDH as a housekeeping protein.

**Figure 5 cells-13-00422-f005:**
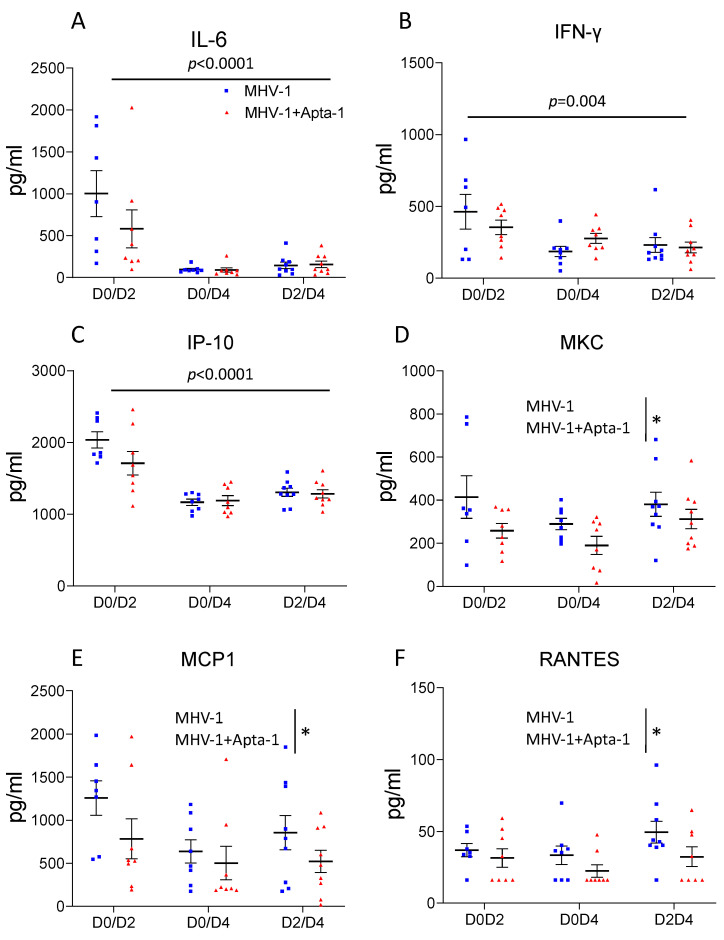
Apta-1 treatment significantly reduce several cytokine levels in the serum. MHV-1 airway infection-induced circulating levels of IL-6 (**A**), IFN-γ (**B**) and IP-10 (**C**) were significantly reduced at Day 4, which were not affected by Apta-1 treatment. The serum levels of MKC (**D**), MCP-1 (**E**), and RANTES (**F**), were significantly lower in Apta-1 treated animals. Two-way ANOVA. Vertical line highlights two experimental conditions. *: *p* < 0.05. Numbers of mice in MHV-1 and MHV-1 + Apta-1 groups are: 7 and 8 in D0D2 groups; 8 and 8 in D0D4 groups; 9 and 9 in D2D4 groups.

**Figure 6 cells-13-00422-f006:**
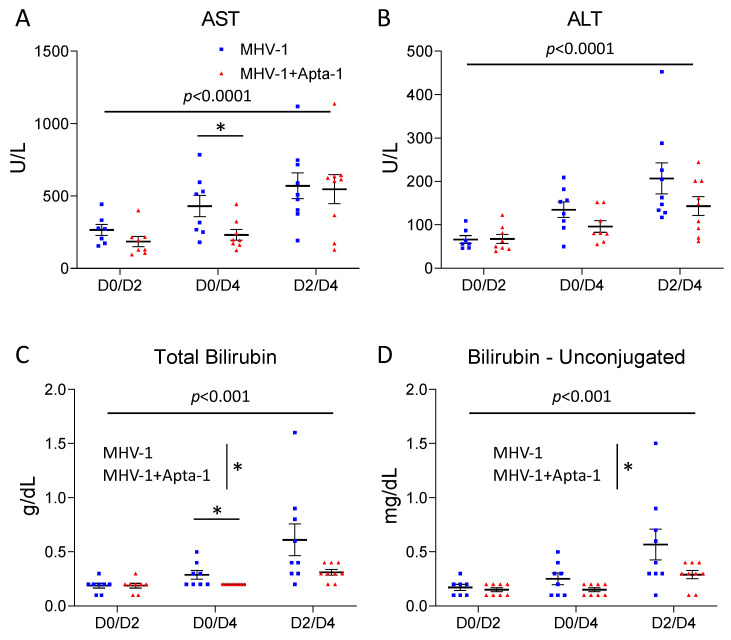
The Apta-1 treatment significantly reduced AST, total bilirubin, and unconjugated bilirubin in the serum. The MHV-1 airway infection induced a significant increase in AST (**A**), ALT (**B**), total bilirubin (**C**), and unconjugated bilirubin (**D**) in the serum at Day 4. The Apta-1 treated animals showed significantly lower levels of AST in the D0D4 group, and significantly lower levels of total bilirubin and unconjugated bilirubin. Two-way ANOVA. The vertical line highlights the two experimental conditions. *: *p* < 0.05. Numbers of mice in MHV-1 and MHV-1 + Apta-1 groups are: 7 and 8 in D0D2 groups; 8 and 8 in D0D4 groups; 9 and 9 in D2D4 groups.

## Data Availability

Data are available upon reasonable request to the corresponding author.

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
