# Peer review of "Therapeutic Effects of a Novel Aptamer on Coronaviral Infection-Induced Lung Injury and Systemic Inflammatory Responses"

_cells, 2024, doi:10.3390/cells13050422_

Round 1

Reviewer 1 Report

Comments and Suggestions for Authors

This manuscript by Dr Wang and colleagues explore the therapeutic potential of an aptamer, APT-1, in pulmonary, viral, infections.   The work appears to be well done and is well presented.   I have a few  concerns related to this manuscript that should be addressed.

1)      My main concern is that, when examining the complete data set, there is no significant difference in all but one of the outcomes in the group where APT-1 was given after 2 days (Group D2D4).  This implies that the majority of the effects of APT-1 were observed in groups where APT-1 was given within 2 hours after viral instillation.  This really limits the conclusion about “therapeutic benefit”; it may still provide important information regarding these infections, but the conclusion regarding therapy may have to be downplayed a bit.   

2)      The statistics need to be clarified. In general, the stats are well described, however there seem to be different tests for different, but similarly designed, experiments. (i.e. some experiments use a t-test as post-hoc, other is Mann Witney U test, etc.    

3)      Few corrections are needed: Line 153 “upper right lung lobes”. There were some issues with symbols (Alpha, Beta etc), this was probably due to the conversion to PDF.  In figure 3. The scale bar is very difficult to see (and it would be better to have a image from each group).

4)      The statement “used mice at 12-14 weeks of age, which are equivalent to older adults in human”, is not really accurate. This mouse age is equivalent to a “mature human” rather than the older group that is susceptible to viral infections.

Comments on the Quality of English Language

minor typographical corrections required

Author Response

Reviewer #1
This manuscript by Dr Wang and colleagues explores the therapeutic potential of an aptamer, APT-1, in pulmonary, viral, infections.   The work appears to be well done and is well presented.   I have a few concerns related to this manuscript that should be addressed.

Response: Thank you for reviewing our manuscript. Please see our responses below.

1)      My main concern is that, when examining the complete data set, there is no significant difference in all but one of the outcomes in the group where APT-1 was given after 2 days (Group D2D4).  This implies that the majority of the effects of APT-1 were observed in groups where APT-1 was given within 2 hours (days) after viral instillation.  This really limits the conclusion about “therapeutic benefit”; it may still provide important information regarding these infections, but the conclusion regarding therapy may have to be downplayed a bit.   

Response: The viral titer was significantly lower in Apta-1 treated animals in D2D4 group (Fig 2C), serum volume was significantly higher in both D0D2 and D0D4 groups (Fig 2E), the hemolysis was significantly lower in D0D4 group (Fig 2F), alveolar hemorrhage was lower in apta-1 treated sample when including all three time groups (Fig 3A), so as reduced PAR1 cleavage (Fig 4B, 4C), reduced IL-6, MKC, MCP1, RANTES (Fig 5A, D, E, F), and lower total bilirubin and unconjugated bilirubin (Fig 6C and 6D) in serum. Therefore, we believe that our conclusion is reasonable. 

2)      The statistics need to be clarified. In general, the stats are well described, however there seem to be different tests for different, but similarly designed, experiments. (i.e. some experiments use a t-test as post-hoc, other is Mann Witney U test, etc.    

Response: When parametric data were passed the normal distribution, we used t-test as a post-hoc test. When non-parametric data (such as semi-quantitative score for lung injury and immunohistochemistry staining) were analyzed, we used Mann Witney U test. 

3)      Few corrections are needed: Line 153 “upper right lung lobes”. There were some issues with symbols (Alpha, Beta etc), this was probably due to the conversion to PDF.  In figure 3. The scale bar is very difficult to see (and it would be better to have a image from each group).

Response: We have corrected the spelling error from “upper right lung robes” to “lobes”.   We also made corrections on alpha, beta etc. We have enhanced the scale bar in Figure 3. H&E photos for other groups were shown in the supplementary figures.

4)      The statement “used mice at 12-14 weeks of age, which are equivalent to older adults in human”, is not really accurate. This mouse age is equivalent to a “mature human” rather than the older group that is susceptible to viral infections.

Response: We have changed the wording as “mature human”.

Reviewer 2 Report

Comments and Suggestions for Authors

See attached file.

Author Response

Reviewer #2
General comments:
The stated objective of this study was to assess the ability of Apta-1, an Aptamer, to mitigate MHV-1 induced acute inflammation and acute lung injury. The results suggest that Apta-1 significantly decreased LPS and MHV-1 induced increases in some of the systemic inflammatory markers, alveolar hemorrhage, and PAR-1 cleavage in the lung. The studies did not include any clinically relevant injury or functional endpoints or cardinal feature of acute lung injury such as airway mechanics, microvascular permeability, lung wet our wet/dry weight, blood oxygenation, heart rate, breathing rate, etc. A more severe injury model than the one chosen would have been more informative regarding the efficacy of Apta-1 against LPS or MHV-1 induced inflammation and acute lung injury.

Response: Since our lung injury data showed limited benefits from Apta-1 treated groups after viral infection, we did not further study functional end points. We have added this critique as one of the limitations of our study in the revised manuscript (page 13, beginning of limitation paragraph). We agree that more severe injury model may be more suitable to elaborate the therapeutic effects of Apta-1. We had discussions in the text.

Specific comments:
-The authors state that they used the LPS model first to identify the local therapeutic effect of Apta-1 in murine lungs before assessing its efficacy against MHV-1 induced inflammation and acute lung injury. However, to do that they used different mice strains (since the A/J strain is expensive) and different ages (6 weeks for the mice used for LPS studies vs. 12-14 weeks for mice used for MHV-1 studies). Not clear why the difference in mice ages for the LPS and MHV-1 studies. Adult mice would have been more appropriate for the LPS studies. No justification for using female mice vs. male or both genders.

Response: For the LPS studies, 6-8 weeks old C57BL/6J mice are commonly used as a model to study inflammatory responses. However, for coronaviral infection, many murine strains are not sensitive to coronaviral strains. As we explained in the Introduction (second paragraph), the use of female AJ mice as a model is through selection of 4 mouse strains and 5 coronavirus strains. We further discussed (page 13, second paragraph) that the use of older mice is to simulate the clinical observation that clinically, older patients showed more severe symptom with worse outcome. Follow previous studies (ref #3, #4), we only used female AJ mice in the present study. We have discussed this as one of the limitations of the present study (page 14, third paragraph).

-The authors should provide the body weights (mean +/- SD or weight range) for the various strains and groups of mice used in this study.

Response: The baseline body weight data of mice (mean +/- SD) were added in the Results.

-Blood collected using a needle through cardiac puncture was used as an index of total blood volume. For a 12–13-week female mouse, the body weight is ~ 22 g and hence the blood volume (~ 75 microliter/g blood) would be ~1,650 microliters. The serum volume would be ~ 50% of the blood volume (~ 850 microliters). Reported serum volumes for control mice are half of the expected numbers. Please report volume of blood collected for the different groups in addition to serum volume and blood hematocrit. Is the change in blood serum collected between the control and MHV-1 group due to dehydration, differences in body weights?

Response: The ~ 75 microliter blood/g body weight is the estimation of total blood volume for normal mice. Practically, only partial of total blood volume can be collected. The blood collection is affected by the anesthesia, the route of collection and many other co-funding factors. Indeed, only about 50% of total blood volume was collected from normal mice. The collection of blood from viral infected mice was much reduced. 
In the present study, the total blood volume was not recorded for every sample. In a subgroup, however, the hematocrit of MHV-1 only group was 50.75 +/-5.84% and 54.61 +/- 3.95% for MHV-1 + Apta-1 group, p>0.05. As shown in Fig 2D, there are no differences of the body weight loss in all three 3 groups, but the serum volume was significantly higher in Apta-1 groups, which reflect the higher collection of total blood.

-No access to supplementary materials. Hence, unable to review such data.

Response: Following the Instruction of the journal publisher, we deposited the Supplementary Material file to Zenodo, and submitted the link to the journal. It appears there is some technical issue. We have contacted Editorial Office to make correction.

-Results, line 200: Why did Apta-1 treatment cause more body weight loss in the LPS model than LPS alone?

Response: As shown in Fig S1A, the body weight loss is about 8% in LPS group and about 12% in LPS+Apat-1 group (p<0.05). Even though it is statistically significant, it is only about 4% difference. On the other hand, in coronaviral infection model, the body weight loss was significantly less in Apta-1 groups. This should be further studied in the future. 

-Results line 261: At what times during day 2 were the 2 doses of Apta-1 administered?

Response: We have changed the sentence as “animals received Apta-1 (100 mg/kg) twice intravenously on Day 2 after viral infection”. Thank you for this comment.

-Results, line 315, “data not shown”: Include data in the supplemental materials.

Response: Following this suggestion, we have added this data in the text.

-Discussion, line 395: Not true for the mice used for the LPS studies

Response: The sentence is about the use of older mice for coronaviral infection model, trying to simulate clinical observations. For LPS studies, the older and younger mice may respond to LPS differently. However, the objective of the present study was on coronaviral infection, not on LPS-induced inflammatory response.